# Global burden and trends of ectopic pregnancy: An observational trend study from 1990 to 2019

Shufei Zhang[☯], Jianfeng Liu[☯], Lian Yang, Hanyue Li, Jianming Tang, Li Hong[ID]*

Department of Obstetrics and Gynecology, Renmin Hospital of Wuhan University, Wuhan, Hubei Province, P. R.C

☯ These authors contributed equally to this work.
* dr_hongli@whu.edu.cn

## Abstract

### Background

Ectopic pregnancy (EP) is one of the leading causes of death in women in early pregnancy, and the mortality of EP have gradually decreased over time in developed countries such as the United Kingdom and the United States. However, epidemiological information on EP has been lacking in recent years, so we analyzed EP data over a thirty-year period from 1990–2019 with the help of Global Burden of Disease study (GBD) data to fill this gap.

### Methods

According to the EP data in GBD for the three decades from 1990 to 2019, we used estimated annual percentage changes (EAPC) to assess the trend of age-standardized incidence rate (ASIR), age-standardized death rate (ASDR) and age-standardized disability adjusted life years (AS-DALYs) trends in EP and to explore the correlation between socio-demographic index (SDI) stratification, age stratification and EP.

### Results

Global ASIR, ASDR, AS-DALYs for EP in 2019 are 170.33/100,000 persons (95% UI: 133.18 to 218.49), 0.16/100,000 persons (95% UI, 0.14 to 0.19) and 9.69/100,000 persons (95% UI, 8.27 to 11.31), respectively. At the overall level, ASDR is significantly negatively correlated with SDI values (R = -0.699, p < 0.001). Besides that, ASDR and AS-DALYs have basically the same pattern. In addition, iron deficiency is one of the risk factors for EP.

### Conclusions

In the past three decades, the morbidity, mortality and disease burden of EP have gradually decreased. It is noteworthy that some economically disadvantaged areas are still experiencing an increase in all indicators, therefore, it is more important to strengthen the protection of women from ethnic minorities and low-income groups.

**Data Availability Statement:** All relevant data are within the manuscript and its Supporting Information files.

**Funding:** This project was funded by The National Key Research and Development Program of China

(2021YFC2701300), Hubei Key Research and Development Program (2022BCA045) and The National Natural Science Foundation of China (81971364 & 82001527). The funders had no role in study design, data collection and analysis, decision to publish, or preparation of the manuscript.

**Competing interests:** The authors have declared that no competing interests exist.

# 1. Introduction

Ectopic pregnancy (EP) is the implantation of the gestational sac outside the uterine cavity and its typical clinical manifestations are menopause, abdominal pain, and vaginal bleeding. And EP is a significant cause of maternal morbidity and unexpected death worldwide [1, 2]. Tubal infection resulting from upper genital tract infection is a major cause of EP, and infectious agents like Mycoplasma genitalium and Chlamydia trachomatis are important risk factors.

Developed countries with well-established healthcare systems have relatively reliable epidemiological data. Previous studies have shown an increase in the incidence of EP in countries like the United States, with the rate rising from 4.5 to 9.4 cases per 1000 reported pregnancies between 1970 and 1978 [3]. By 1989, there was a fourfold increase compared to 1970 [4]. Not only this, but other developed countries such as Canada, New Zealand, and the United Kingdom have shown similar increasing trends in the incidence of EP [5–7]. On the contrary, developing countries, particularly in Africa, have limited epidemiological data on EP, and only a few early studies indicate an increasing trend in EP incidence [8]. A study conducted in China showed that the prevalence of EP was around 2.5% in 2004 and exhibited an overall decreasing trend from 2011 to 2020. However, a change in fertility policy in 2015 resulted in an increase in the proportion of EP among individuals aged 35 years and older [9]. It is worrisome that underdeveloped countries lack comprehensive epidemiological data on EP due to poor medical and economic conditions. Consequently, assessing the prevalence of risk factors such as Chlamydia trachomatis infection and pelvic inflammatory disease is the only way to speculate, making the EP situation in underdeveloped countries less optimistic [10].

Despite a significant decrease in EP mortality in countries like the United Kingdom and the United States at the end of the last century, it remains high in developing countries [1]. The incidence of EP continues to rise, resulting in a substantial disease burden. However, global epidemiological studies of EP are still lacking.

Explore results from the 2019 Global Burden of Disease study (GBD 2019), which published epidemiological data related to 369 diseases/injuries and 286 causes of death, covered EP in 204 countries and territories from 1990 to 2019 [11]. In this context, we analyzed the GBD data from 1990 to 2019, examining various aspects such as incidence, mortality, and risk factors associated with EP, aiming to support the management of EP patients worldwide and inform public health policy development.

# 2. Materials and methods

## 2.1 Data acquisition

In GBD, EP is defined as pregnancy occurring outside of the uterus. (https://www.healthdata.org/results/gbd_summaries/2019/ectopic-pregnancy-level-4-cause). Study data was obtained from GBD 2019 was modeled by the Institute for Health Metrics and Evaluation (IHME) [12].

The Socio-demographic Index (SDI) is a composite indicator of development status strongly correlated with health outcomes. It is the geometric mean of 0 to 1 indices of total fertility rate under the age of 25, mean education for those ages 15 and older and lag distributed income per capita. As a composite, a location with an SDI of 0 would have a theoretical minimum level of development relevant to health, while a location with an SDI of 1 would have a theoretical maximum level [13] (https://ghdx.healthdata.org/gbd-2019). SDI data was obtained from Global Health Data Exchange (GHDx).

## 2.2 Statistical analysis

To evaluate trends in incidence rate, death rate, and burden of EP, we calculated relevant assessment indicators, namely annual age-standardized incidence rate (ASIR), age-standardized mortality rate (ASDR), and age-standardized DALYs rate (AS-DALYs). Not only that, we used annual percentage change (EAPC) to accurately evaluate the trend of ASR [14].

The EAPC is calculated by fitting the linear regression line: Y = α + βx + ε, where y represents ln(ASR) and x refers to the calendar year. The value of EAPC equals 100 × (exp(β) − 1) and its 95% CI is attainable in the regression model [15, 16]. And the ASR was obtained as follows:

$$ASR = \frac{\sum_{i=1}^{A} aiwi}{\sum_{i=1}^{A} wi} \times 10000$$

In the $i$th age subgroup, ai is represented as age class. $wi$ denotes the number of persons (or weight), where $i$ is equal to the selected reference standard population [17]. when both the EAPC value and its 95% CI >0, we consider its ASR to be on an upward trend; when both the EAPC value and 95% CI <0, we consider its ASR to be on an downward trend; In other cases, we consider the ASR to be stable [18]. We use Pearson's correlation coefficient (R) to represent the strength of the correlation, and all analyses and data visualization are done in R software (version 4.2.1, http://www.r-project.org/, Auckland, New Zealand).

## 3. Results

### 3.1 Distribution and trends in the incidence rate of EP

At the global level, there were 6.7 million (95% UI: 5.2 to 8.6) incident cases of EP in 2019, with an ASIR of 170.33/100,000 persons (95% UI: 133.18 to 218.49). The number of cases in 2019 was 0.1% lower than in 1990 (95% UI: -0.16 to -0.04). It is worth drawing our attention to the fact that only the number of incidence cases in the low SDI region increased by 0.53% (95% UI: 0.48 to 0.58) during these three decades (Table 1). On observation from the GBD regions and countries level, the three countries with the highest ASIR are Niger, Papua New Guinea, and Chad; the three countries with the lowest ASIR are Australia, South Africa, and Poland (Fig 1A and S1 Table).

In the global overall level analysis, incidence rates were lower in 2019 than in 1990 for all age stages, with peak incidence rates occurring in the 25–29 age group. incidence rates were significantly higher in women aged 25–34 years, which also appears to be consistent with a sexually active period for women. incidence rates in 2019 were generally lower than 1990 incidence rates, and interestingly, incidence rates in high SDI regions in the 30+ stage exceeded those in 1990, while incidence rates in high-moderate SDI regions exceeded those in 1990 in the 35+ stage. Not only that, the peak prevalence in each region was largely consistent with the global level, with the prevalence peaking at 25–29 years of age in both 1990 and 2019, while the high SDI region showed a pushed-back peak in 2019 (Fig 2).

On observation from the GBD regions and countries level, the three countries with the highest Case changes are Qatar, Afghanistan, and Somalia; the three countries with the lowest Case changes are the Northern Mariana Islands, Albania, and Puerto Rico (S1 Fig). And the three countries with the highest EAPC are Russian Federation, Italy, and Belarus; the three countries with the lowest EAPC are Oman, Nepal, and Kuwait (Fig 1D).

### 3.2 Distribution and trends in the DALYs rate of EP

At the global level, there were 0.34 million (95% UI: 0.30–0.38) DALYs in 1990 and 0.38 million (95% UI: 0.32–0.44) DALYs in 2019. In the past 30 years, the AS-DALYs rate decreased

**Table 1. Incidence of ectopic pregnancy in 1990 and 2019 for all locations, with EAPC from 1990 and 2019.**

| Location | 1990 | | 2019 | | 1990–2019 | |
|---|---|---|---|---|---|---|
| | Incident cases No. × 10³(95% UI) | ASR per 100,000 No. (95% UI) | Incident cases No. × 10³(95% UI) | ASR per 100,000 No. (95% UI) | Case change No. (95% UI) | EAPC No. (95% CI) |
| Global | 7453.3 (5739 to 9557.1) | 266.79 (204.3 to 343) | 6692.4 (5225.4 to 8598.6) | 170.33 (133.18 to 218.49) | -0.1% (-0.16 to -0.04) | -1.15 (-1.31 to -0.99) |
| **Socio-demographic index** | | | | | | |
| Low | 983.2 (765.5 to 1292.7) | 418.6 (326.44 to 548.26) | 1501.7 (1156 to 1982.9) | 273.81 (211.25 to 359.81) | 0.53% (0.48 to 0.58) | -1.47 (-1.57 to -1.38) |
| Low-middle | 1778.5 (1369.1 to 2281.4) | 322.26 (249.36 to 413.27) | 1512.4 (1160.3 to 1976.9) | 158.49 (122.09 to 207.04) | -0.15% (-0.19 to -0.09) | -2.29 (-2.33 to -2.24) |
| Middle | 2269 (1717.2 to 2955.8) | 242.8 (182.99 to 319.2) | 1659.3 (1267 to 2189.6) | 130.52 (99.99 to 170.65) | -0.27% (-0.34 to -0.2) | -1.59 (-1.85 to -1.33) |
| Middle-high | 1795.2 (1362.1 to 2341.2) | 285.13 (216.49 to 372.19) | 1461.7 (1121.1 to 1931.8) | 199.34 (153.88 to 260.35) | -0.19% (-0.28 to -0.08) | -0.31 (-0.69 to 0.08) |
| High | 624.3 (466.7 to 833.9) | 144.76 (108.1 to 191.81) | 553.2 (434.8 to 705) | 119.41 (94.75 to 153.59) | -0.11% (-0.22 to 0.01) | -0.66 (-0.72 to -0.6) |
| **Region** | | | | | | |
| High-income Asia Pacific | 64.2 (46.1 to 88) | 74.71 (53.42 to 102.65) | 46 (35.3 to 60.6) | 61.66 (47.63 to 81.69) | -0.28% (-0.38 to -0.14) | -0.72 (-0.9 to -0.53) |
| Central Asia | 118.5 (88 to 156.6) | 319.83 (239.91 to 420.44) | 117.1 (86.8 to 154.9) | 230.1 (170.92 to 302.57) | -0.01% (-0.07 to 0.06) | -0.65 (-0.99 to -0.31) |
| East Asia | 2557 (1898.9 to 3370.1) | 356.51 (263.65 to 473.15) | 1463 (1108.3 to 1970.1) | 192.01 (145.74 to 253.91) | -0.43% (-0.51 to -0.34) | -0.87 (-1.36 to -0.39) |
| South Asia | 1562.5 (1192.9 to 2072.7) | 292.68 (224.78 to 387.73) | 1351.9 (992.4 to 1808.8) | 136.56 (101.21 to 182.39) | -0.13% (-0.19 to -0.06) | -2.68 (-2.72 to -2.64) |
| Southeast Asia | 404.4 (307.7 to 534.5) | 164.93 (125.99 to 219.09) | 377.8 (289 to 506.8) | 104.78 (79.97 to 140.16) | -0.07% (-0.12 to -0.02) | -1.44 (-1.49 to -1.39) |
| Australasia | 9 (6.3 to 13.2) | 82.39 (57.81 to 119.74) | 9.1 (7.1 to 11.7) | 65.74 (51.75 to 84.72) | 0.01% (-0.19 to 0.26) | -0.6 (-0.95 to -0.25) |
| Caribbean | 34.4 (25.9 to 45.6) | 179.56 (135.15 to 238.74) | 34.1 (25.6 to 45.8) | 140.35 (105.88 to 188.53) | -0.01% (-0.08 to 0.07) | -0.78 (-0.87 to -0.7) |
| Central Europe | 81.8 (61.8 to 107.3) | 143.21 (107.21 to 188.92) | 57.7 (44.4 to 74.5) | 117.39 (89.64 to 151.53) | -0.3% (-0.4 to -0.17) | -0.35 (-0.71 to 0.02) |
| Eastern Europe | 360.8 (273.2 to 477) | 329.52 (251.34 to 437.56) | 350.2 (260.9 to 469.6) | 349.96 (262.95 to 457.78) | -0.03% (-0.18 to 0.14) | 1.44 (0.85 to 2.04) |
| Western Europe | 311.6 (233.2 to 415.3) | 160.01 (119.86 to 212.77) | 341.3 (259.1 to 444.1) | 181.16 (137.85 to 238.2) | 0.1% (-0.03 to 0.25) | 0.68 (0.58 to 0.79) |
| Andean Latin America | 75.9 (58.5 to 98.6) | 404.47 (310.72 to 534.09) | 89.5 (68.1 to 117.3) | 265.71 (202.64 to 347.58) | 0.18% (0.09 to 0.28) | -1.26 (-1.4 to -1.13) |
| Central Latin America | 195.8 (145.6 to 264.7) | 232.32 (172.32 to 313.57) | 207.1 (161.7 to 270.5) | 152.73 (119.06 to 199.48) | 0.06% (-0.02 to 0.15) | -1.88 (-2.05 to -1.71) |
| Southern Latin America | 112.3 (84.5 to 148.4) | 448.06 (336.78 to 590.45) | 117 (87 to 158.2) | 336.75 (250.85 to 450.64) | 0.04% (-0.06 to 0.17) | -0.83 (-0.93 to -0.73) |
| Tropical Latin America | 99.1 (76.2 to 129.7) | 118.85 (91.56 to 155.51) | 103.5 (77.7 to 140.5) | 85.28 (64.24 to 115.29) | 0.04% (-0.04 to 0.13) | -0.97 (-1.08 to -0.86) |
| North Africa and Middle East | 389.1 (300.2 to 510.7) | 246.28 (190.51 to 324.35) | 443.4 (338.2 to 595.1) | 134.89 (103.09 to 180.11) | 0.14% (0.07 to 0.22) | -2.01 (-2.13 to -1.88) |
| High-income North America | 194.7 (137.3 to 269) | 127.15 (90.15 to 175.8) | 132.5 (111.1 to 162.9) | 79.81 (66.99 to 98.02) | -0.32% (-0.45 to -0.13) | -2.36 (-2.78 to -1.94) |
| Oceania | 13 (9.7 to 17.4) | 429.24 (321.55 to 568.66) | 25.4 (19 to 33.7) | 377.18 (281.64 to 502.22) | 0.96% (0.78 to 1.15) | -0.46 (-0.47 to -0.44) |
| Central Sub-Saharan Africa | 105.5 (81 to 141.5) | 431.16 (326.68 to 572.8) | 175.6 (132.1 to 237.7) | 282.28 (211.09 to 383.12) | 0.66% (0.53 to 0.86) | -1.31 (-1.5 to -1.11) |

(*Continued*)

**Table 1.** (Continued)

| Location | 1990 | | 2019 | | 1990–2019 | |
|---|---|---|---|---|---|---|
| | Incident cases No. × 10³(95% UI) | ASR per 100,000 No. (95% UI) | Incident cases No. × 10³(95% UI) | ASR per 100,000 No. (95% UI) | Case change No. (95% UI) | EAPC No. (95% CI) |
| **Eastern Sub-Saharan Africa** | 359.1 (276.4 to 473.6) | 423.25 (326.19 to 560.24) | 550.5 (421.7 to 735.7) | 270.41 (206.59 to 360.42) | 0.53% (0.47 to 0.59) | -1.57% (-1.65 to -1.49) |
| **Southern Sub-Saharan Africa** | 35.3 (27.3 to 45.8) | 127.56 (98.91 to 165.69) | 36 (27.7 to 46.9) | 81.41 (62.94 to 105.76) | 0.02% (-0.05 to 0.09) | -1.49% (-1.6 to -1.39) |
| **Western Sub-Saharan Africa** | 369.1 (286.3 to 486) | 431.56 (334.89 to 570.4) | 663.8 (509.5 to 884.7) | 299.34 (229.06 to 398.25) | 0.8% (0.75 to 0.85) | -1.21% (-1.3 to -1.11) |

with an EAPC of -0.84% (95%CI: from -0.98 to -0.7), dropping from 12.46/100,000 persons (95% UI, 11.09 to 13.91) in 1990 to 9.69/100,000 persons (95% UI, 8.27 to 11.31) in 2019. Over the past three decades, DALYs in middle SDI to high SDI regions have become lower and maintained the highest EAPC of -4.53 (-4.75 to -4.3) with a clear downward trend, while DALYs in low and low-middle SDI regions have not only not decreased but even slightly increased, not only that, the downward trend of EAPC values is not obvious. Looking at the global regions, DALYs generally decreased, but the four regions in Sub-Saharan Africa had the highest AS-DALYs at 60.76/100,000 (44.5 to 77.37). While the Caribbean had the most pronounced increase at 4.45% (3.07 to 6.12), EAPC also had the most pronounced upward trend at 6.11% (5.22 to 7.02) (Table 2). On observation from the GBD regions and countries level, the three countries with the highest AS-DALYs are Chad, Mauritania, and Senegal; the three countries with the lowest AS-DALYs are Poland, Singapore, Cyprus (Fig 1B and S2 Table). The age distribution pattern of DALYs rate in most regions was largely consistent with that of death rate (S2 and S3 Figs).

### 3.3 Distribution and trends in the death rate of EP

As we mentioned earlier, EP does not cause serious consequences if detected in time, but rupture of EP is a gynecological emergency that can lead to death in women [18]. Globally, there were 5749 (95% UI: 5107 to 6435) deaths from EP in 1990 and 6452 (95% UI: 5496 to 7513) deaths from EP in 2019. Over the last 30 years, ASDR has decreased and continues to show a downward trend from 0.22/100,000 (95% UI, 0.19 to 0.24) in 1990 decreasing to 0.16/100,000 (95% UI, 0.14 to 0.19) in 2019, with an EAPC of -0.91% (95% CI: -1.04 to -0.78).

In terms of SDI, lower SDI does seem to lead to higher ASDR, and this difference can even be tens of times higher. 2019 ASDR in low SDI areas was 0.68 (0.56 to 0.82), while ASDR in high SDI areas was only 0.01 (0.01 to 0.01), and EAPC was higher at -2.95% (-3.13 to– 2.77), suggesting a more pronounced downward trend in high SDI regions along with a decrease in mortality (Table 3).

Looking at the GBD region and country level, the three countries with the highest ASDR were Mauritania, Chad, and Senegal; earlier detection of EP did not result in death, so the lowest countries were not listed (Fig 1C and S3 Table). Unlike acute diseases such as acute myocardial infarction, EP has a relatively low lethality rate. Globally, mortality from EP has declined after three decades compared with 1990, but the pattern of its age distribution is essentially the same as before (S2 Fig). By combining morbidity and mortality data, we performed a cluster analysis at the national and regional levels to find countries with similar annual increases. Based on the results of the cluster analysis, 59 countries (or regions) were classified in the "Significant increase" group, including the Netherlands, China, the United Kingdom, and Germany. 90 countries (or regions) were classified in the "Minor increase"

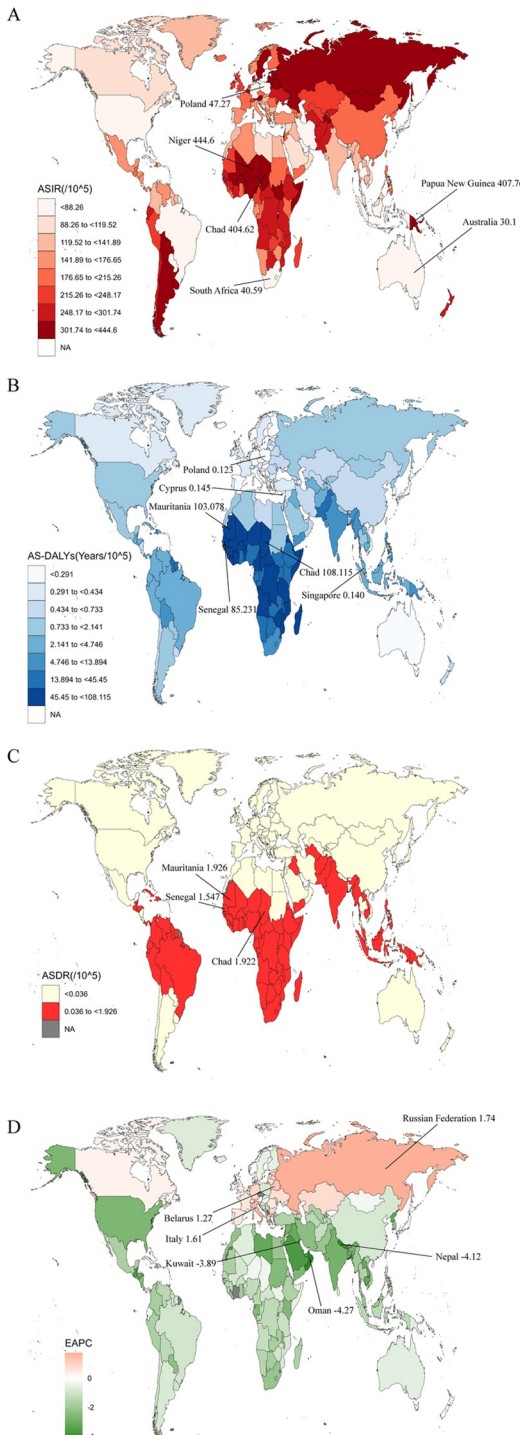

**Fig 1.** Distribution of (A) ASIR, (B) ASDR, (C) AS-DALYs and (D) EAPC-ASIR in various countries and regions in 2019.

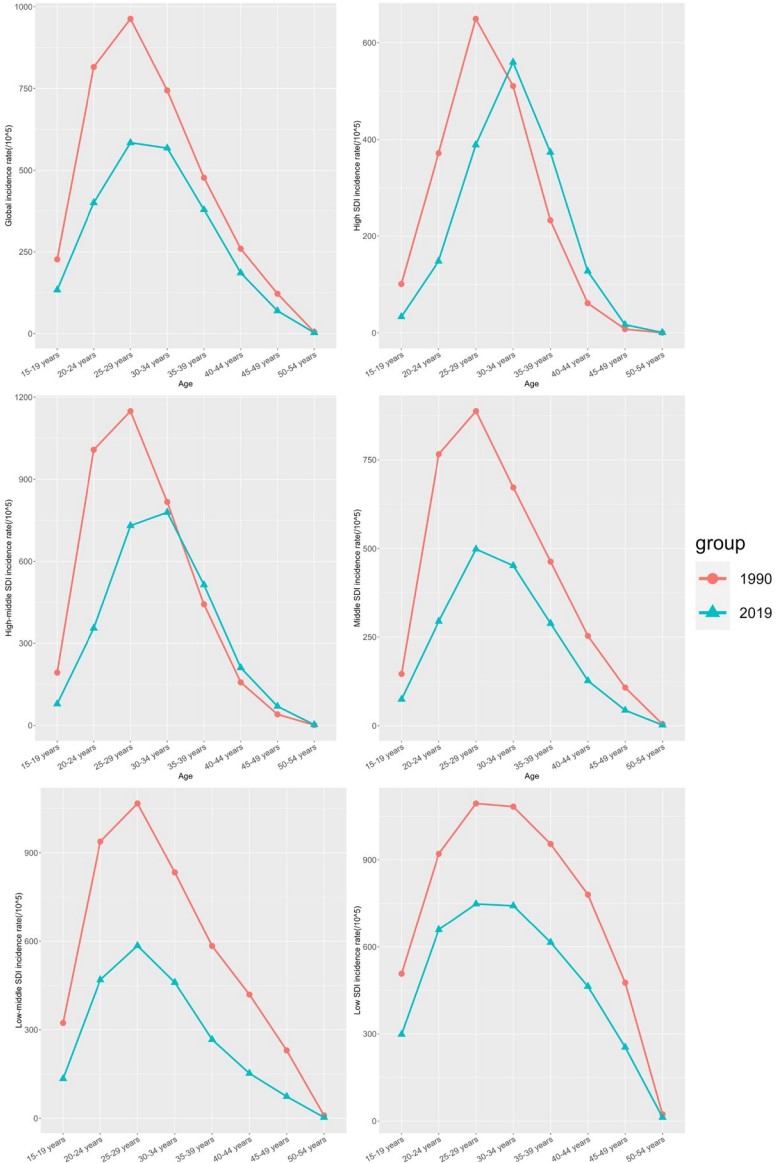

**Fig 2. Distribution of incidence rate in different SDI regions, age distribution of incidence rate in different SDI regions from 1990–2019.**

group, including Thailand, Mexico, Egypt, and the Republic of Korea. 24 countries (or regions) were classified in the "Stable or minor decrease" group, including Canada, Cuba, Niger, and Chad. The remaining 39 countries (or regions) are classified in the "Significant decrease" group, including Oman, Sudan, Libya, and Yemen (Fig 3).

## 3.4 Correlation analysis of EP related ASIR, ASDR, AS-DALYs and different SDI

In 2019, a significant association was detected between EAPC and ASIR, ($\rho = 0.34$, $p < 0.001$), and similarly, a significant association was observed between EAPC and ASDR, ($\rho = 0.29$, $p < 0.001$) (Fig 4A). We can observe the ASIR and its expected levels for different SDI regions

**Table 2.** DALYs of ectopic pregnancy in 1990 and 2019 for all locations, with EAPC from 1990 and 2019.

| Location | 1990 | | 2019 | | 1990–2019 | |
|---|---|---|---|---|---|---|
| | DALYs No. × 10³(95% UI) | AS-DALY per 100,000 No. (95% UI) | DALYs No. × 10³(95% UI) | AS-DALY per 100,000 No. (95% UI) | DALYs change No. (95% UI) | EAPC No. (95% CI) |
| Global | 339.44 (301.77 to 378.65) | 12.46 (11.09 to 13.91) | 378.03 (322.55 to 440.09) | 9.69 (8.27 to 11.31) | 0.11% (-0.07 to 0.31) | -0.84% (-0.98 to -0.7) |
| **Socio-demographic index** | | | | | | |
| Low | 115.84 (98.28 to 134.88) | 51.46 (43.7 to 59.7) | 200.33 (164.29 to 239.91) | 37.69 (30.83 to 45.33) | 0.73% (0.39 to 1.1) | -0.97% (-1.11 to -0.82) |
| Low-middle | 116.16 (99.96 to 134.1) | 21.57 (18.63 to 24.91) | 124.92 (104.84 to 146.25) | 13.21 (11.09 to 15.47) | 0.08% (-0.11 to 0.3) | -1.6% (-1.83 to -1.37) |
| Middle | 73.43 (65.37 to 81.58) | 8.04 (7.16 to 8.95) | 41.13 (35.01 to 48.43) | 3.31 (2.82 to 3.91) | -0.44% (-0.54 to -0.33) | -3.18% (-3.51 to -2.84) |
| Middle-high | 28.78 (25.58 to 32.24) | 4.63 (4.12 to 5.19) | 9.1 (7.81 to 10.63) | 1.32 (1.14 to 1.55) | -0.68% (-0.73 to -0.62) | -4.53% (-4.75 to -4.3) |
| High | 5.12 (4.55 to 5.75) | 1.19 (1.06 to 1.34) | 2.33 (1.99 to 2.73) | 0.52 (0.44 to 0.6) | -0.54% (-0.61 to -0.48) | -2.62% (-2.78 to -2.46) |
| **Region** | | | | | | |
| High-income Asia Pacific | 0.63 (0.55 to 0.73) | 0.71 (0.62 to 0.82) | 0.14 (0.11 to 0.17) | 0.19 (0.16 to 0.23) | -0.78% (-0.82 to -0.73) | -4.75% (-4.97 to -4.53) |
| Central Asia | 0.48 (0.41 to 0.56) | 1.3 (1.12 to 1.52) | 0.37 (0.3 to 0.45) | 0.73 (0.59 to 0.89) | -0.23% (-0.34 to -0.11) | -1.9% (-2.08 to -1.72) |
| East Asia | 27.22 (22.16 to 33.14) | 3.83 (3.12 to 4.68) | 5.08 (3.99 to 6.27) | 0.7 (0.55 to 0.87) | -0.81% (-0.86 to -0.75) | -5.73% (-6.17 to -5.29) |
| South Asia | 121.3 (98.97 to 146.95) | 23.21 (19.09 to 28.03) | 106.31 (83.76 to 131.84) | 10.87 (8.56 to 13.47) | -0.12% (-0.34 to 0.15) | -2.86% (-3.16 to -2.56) |
| Southeast Asia | 19.15 (16.29 to 22.64) | 7.87 (6.68 to 9.31) | 13.88 (11.44 to 16.41) | 3.86 (3.19 to 4.56) | -0.28% (-0.43 to -0.08) | -2.69% (-2.94 to -2.43) |
| Australasia | 0.08 (0.07 to 0.1) | 0.75 (0.61 to 0.91) | 0.04 (0.03 to 0.05) | 0.28 (0.22 to 0.35) | -0.54% (-0.65 to -0.39) | -3.35% (-4.03 to -2.66) |
| Caribbean | 0.55 (0.45 to 0.67) | 2.83 (2.35 to 3.46) | 2.97 (2.22 to 3.88) | 12.24 (9.12 to 15.95) | 4.45% (3.07 to 6.12) | 6.11% (5.22 to 7.02) |
| Central Europe | 0.78 (0.71 to 0.86) | 1.29 (1.17 to 1.43) | 0.12 (0.1 to 0.16) | 0.25 (0.2 to 0.32) | -0.84% (-0.87 to -0.8) | -5.86% (-6.44 to -5.28) |
| Eastern Europe | 6.82 (5.82 to 7.91) | 5.89 (5.02 to 6.86) | 0.81 (0.61 to 1.05) | 0.81 (0.61 to 1.05) | -0.88% (-0.91 to -0.84) | -7.64% (-8 to -7.29) |
| Western Europe | 1.39 (1.22 to 1.59) | 0.71 (0.63 to 0.82) | 0.57 (0.42 to 0.77) | 0.31 (0.23 to 0.41) | -0.59% (-0.67 to -0.5) | -3.05% (-3.21 to -2.89) |
| Andean Latin America | 0.54 (0.45 to 0.66) | 2.74 (2.29 to 3.32) | 1.73 (1.29 to 2.27) | 5.16 (3.85 to 6.76) | 2.22% (1.34 to 3.45) | 2.07% (1.06 to 3.09) |
| Central Latin America | 4.33 (3.89 to 4.83) | 4.91 (4.42 to 5.47) | 3.27 (2.69 to 4.03) | 2.41 (1.98 to 2.96) | -0.25% (-0.39 to -0.06) | -2.51% (-2.83 to -2.2) |
| Southern Latin America | 0.38 (0.31 to 0.46) | 1.52 (1.26 to 1.84) | 0.5 (0.41 to 0.6) | 1.47 (1.22 to 1.77) | 0.31% (0.07 to 0.62) | -0.05% (-0.39 to 0.3) |
| Tropical Latin America | 5.61 (4.76 to 6.59) | 6.89 (5.88 to 8.05) | 2.79 (2.36 to 3.3) | 2.32 (1.97 to 2.75) | -0.5% (-0.61 to -0.37) | -2.62% (-3.34 to -1.9) |
| North Africa and Middle East | 5.75 (5.06 to 6.49) | 3.6 (3.16 to 4.07) | 4.51 (3.66 to 5.51) | 1.39 (1.13 to 1.7) | -0.22% (-0.36 to -0.04) | -3.33% (-3.42 to -3.25) |
| High-income North America | 2.67 (2.24 to 3.14) | 1.78 (1.49 to 2.09) | 1.28 (1.05 to 1.54) | 0.78 (0.64 to 0.94) | -0.52% (-0.62 to -0.4) | -2.4% (-2.71 to -2.1) |
| Oceania | 0.42 (0.33 to 0.52) | 13.56 (10.76 to 16.61) | 0.8 (0.61 to 1.04) | 11.75 (8.98 to 15.26) | 0.89% (0.45 to 1.5) | -0.59% (-0.86 to -0.32) |
| Central Sub-Saharan Africa | 16.6 (12.59 to 21.08) | 69.68 (52.38 to 88.9) | 36.76 (26.7 to 46.19) | 60.76 (44.5 to 77.37) | 1.21% (0.56 to 2.12) | 0.29% (-0.03 to 0.61) |

*(Continued)*

**Table 2.**  (Continued)

| Location | 1990 | | 2019 | | 1990–2019 | |
|---|---|---|---|---|---|---|
| | DALYs<br>No. × 10³(95% UI) | AS-DALY<br>per 100,000<br>No. (95% UI) | DALYs<br>No. × 10³(95% UI) | AS-DALY<br>per 100,000<br>No. (95% UI) | DALYs change<br>No. (95% UI) | EAPC<br>No. (95% CI) |
| **Eastern Sub-Saharan Africa** | 46.52 (38.46 to 54.78) | 61.49 (51 to 72.62) | 72.11 (58.13 to 87.87) | 38.69 (31.14 to 46.9) | 0.55% (0.19 to 0.92) | -1.47% (-1.59 to -1.35) |
| **Southern Sub-Saharan Africa** | 15.26 (12.93 to 17.87) | 56.75 (47.99 to 66.42) | 9.78 (7.26 to 12.69) | 22.16 (16.56 to 28.63) | -0.36% (-0.54 to -0.14) | -1.83% (-2.95 to -0.7) |
| **Western Sub-Saharan Africa** | 62.98 (50.41 to 79.57) | 75.16 (60.23 to 94.67) | 114.2 (88.17 to 148.99) | 51.49 (39.68 to 66.93) | 0.81% (0.36 to 1.39) | -1.28% (-1.54 to -1.01) |

**Table 3. Deaths of ectopic pregnancy in 1990 and 2019 for all locations, with EAPC from 1990 and 2019.**

| Location | 1990 | | 2019 | | 1990–2019 | |
|---|---|---|---|---|---|---|
| | Death cases<br>No. (95% UI) | ASDR<br>per 100,000<br>No. (95% UI) | Death cases<br>No. (95% UI) | ASDR<br>per 100,000<br>No. (95% UI) | Case change<br>No. (95% UI) | EAPC<br>No. (95% CI) |
| **Global** | 5749 (5107 to 6435) | 0.22 (0.19 to 0.24) | 6452 (5496 to 7513) | 0.16 (0.14 to 0.19) | 0.12% (-0.06 to 0.32) | -0.91% (-1.04 to -0.78) |
| **Socio-demographic index** | | | | | | |
| **Low** | 2032 (1723 to 2359) | 0.94 (0.79 to 1.09) | 3478 (2849 to 4187) | 0.68 (0.56 to 0.82) | 0.71% (0.38 to 1.09) | -1.02% (-1.15 to -0.89) |
| **Low-middle** | 1968 (1698 to 2279) | 0.38 (0.33 to 0.44) | 2126 (1778 to 2495) | 0.23 (0.19 to 0.27) | 0.08% (-0.11 to 0.3) | -1.69% (-1.88 to -1.49) |
| **Middle** | 1209 (1075 to 1345) | 0.14 (0.12 to 0.15) | 679 (575 to 802) | 0.05 (0.05 to 0.06) | -0.44% (-0.54 to -0.32) | -3.3% (-3.63 to -2.96) |
| **Middle-high** | 460 (412 to 514) | 0.07 (0.07 to 0.08) | 134 (115 to 155) | 0.02 (0.02 to 0.02) | -0.71% (-0.76 to -0.65) | -4.93% (-5.19 to -4.67) |
| **High** | 78 (70 to 87) | 0.02 (0.02 to 0.02) | 32 (28 to 36) | 0.01 (0.01 to 0.01) | -0.59% (-0.65 to -0.52) | -2.95% (-3.13 to -2.77) |
| **Region** | | | | | | |
| **High-income Asia Pacific** | 10 (9 to 12) | 0.01 (0.01 to 0.01) | 2 (1 to 2) | 0 (0 to 0) | -0.83% (-0.86 to -0.79) | -5.67% (-5.88 to -5.45) |
| **Central Asia** | 6 (6 to 7) | 0.02 (0.02 to 0.02) | 4 (4 to 5) | 0.01 (0.01 to 0.01) | -0.29% (-0.41 to -0.14) | -2.37% (-2.52 to -2.23) |
| **East Asia** | 414 (332 to 512) | 0.06 (0.05 to 0.07) | 65 (49 to 82) | 0.01 (0.01 to 0.01) | -0.84% (-0.89 to -0.78) | -6.45% (-6.96 to -5.93) |
| **South Asia** | 2040 (1673 to 2471) | 0.4 (0.33 to 0.49) | 1778 (1397 to 2212) | 0.18 (0.14 to 0.23) | -0.13% (-0.35 to 0.15) | -2.95% (-3.22 to -2.69) |
| **Southeast Asia** | 323 (273 to 386) | 0.14 (0.12 to 0.16) | 234 (192 to 276) | 0.06 (0.05 to 0.08) | -0.28% (-0.43 to -0.07) | -2.81% (-3.08 to -2.54) |
| **Australasia** | 1 (1 to 2) | 0.01 (0.01 to 0.01) | 1 (0 to 1) | 0 (0 to 0) | -0.59% (-0.7 to -0.44) | -3.89% (-4.69 to -3.08) |
| **Caribbean** | 9 (7 to 11) | 0.05 (0.04 to 0.06) | 51 (37 to 67) | 0.21 (0.15 to 0.27) | 4.9% (3.35 to 6.77) | 6.33% (5.41 to 7.27) |
| **Central Europe** | 12 (11 to 14) | 0.02 (0.02 to 0.02) | 1 (1 to 1) | 0 (0 to 0) | -0.9% (-0.92 to -0.88) | -7.79% (-8.32 to -7.26) |
| **Eastern Europe** | 112 (96 to 130) | 0.1 (0.08 to 0.11) | 8 (6 to 11) | 0.01 (0.01 to 0.01) | -0.92% (-0.95 to -0.9) | -9.34% (-9.78 to -8.9) |
| **Western Europe** | 19 (18 to 21) | 0.01 (0.01 to 0.01) | 4 (4 to 5) | 0 (0 to 0) | -0.77% (-0.8 to -0.74) | -5.24% (-5.55 to -4.92) |
| **Andean Latin America** | 8 (6 to 9) | 0.04 (0.03 to 0.05) | 28 (20 to 37) | 0.08 (0.06 to 0.11) | 2.6% (1.55 to 4.06) | 2.32% (1.23 to 3.43) |
| **Central Latin America** | 69 (62 to 76) | 0.08 (0.07 to 0.09) | 51 (42 to 64) | 0.04 (0.03 to 0.05) | -0.25% (-0.41 to -0.06) | -2.62% (-2.94 to -2.29) |
| **Southern Latin America** | 5 (4 to 6) | 0.02 (0.02 to 0.02) | 7 (5 to 8) | 0.02 (0.02 to 0.02) | 0.37% (0.09 to 0.72) | -0.04% (-0.48 to 0.41) |
| **Tropical Latin America** | 95 (80 to 111) | 0.12 (0.1 to 0.14) | 46 (39 to 54) | 0.04 (0.03 to 0.04) | -0.52% (-0.62 to -0.38) | -2.78% (-3.54 to -2.02) |
| **North Africa and Middle East** | 92 (80 to 103) | 0.06 (0.05 to 0.07) | 71 (57 to 89) | 0.02 (0.02 to 0.03) | -0.22% (-0.37 to -0.04) | -3.43% (-3.52 to -3.34) |
| **High-income North America** | 42 (35 to 50) | 0.03 (0.02 to 0.03) | 20 (16 to 24) | 0.01 (0.01 to 0.01) | -0.53% (-0.63 to -0.4) | -2.33% (-2.65 to -2.01) |
| **Oceania** | 7 (6 to 9) | 0.23 (0.18 to 0.29) | 13 (10 to 18) | 0.2 (0.15 to 0.26) | 0.92% (0.46 to 1.54) | -0.61% (-0.88 to -0.33) |
| **Central Sub-Saharan Africa** | 289 (217 to 369) | 1.26 (0.94 to 1.61) | 643 (468 to 816) | 1.1 (0.8 to 1.42) | 1.23% (0.56 to 2.16) | 0.29% (-0.02 to 0.61) |
| **Eastern Sub-Saharan Africa** | 848 (704 to 1000) | 1.18 (0.98 to 1.39) | 1297 (1044 to 1579) | 0.73 (0.59 to 0.88) | 0.53% (0.17 to 0.89) | -1.54% (-1.65 to -1.43) |
| **Southern Sub-Saharan Africa** | 262 (222 to 308) | 1 (0.85 to 1.17) | 168 (125 to 217) | 0.38 (0.29 to 0.49) | -0.36% (-0.54 to -0.15) | -1.95% (-3.05 to -0.83) |
| **Western Sub-Saharan Africa** | 1086 (869 to 1373) | 1.35 (1.08 to 1.69) | 1959 (1511 to 2554) | 0.92 (0.7 to 1.2) | 0.8% (0.35 to 1.37) | -1.33% (-1.59 to -1.08) |

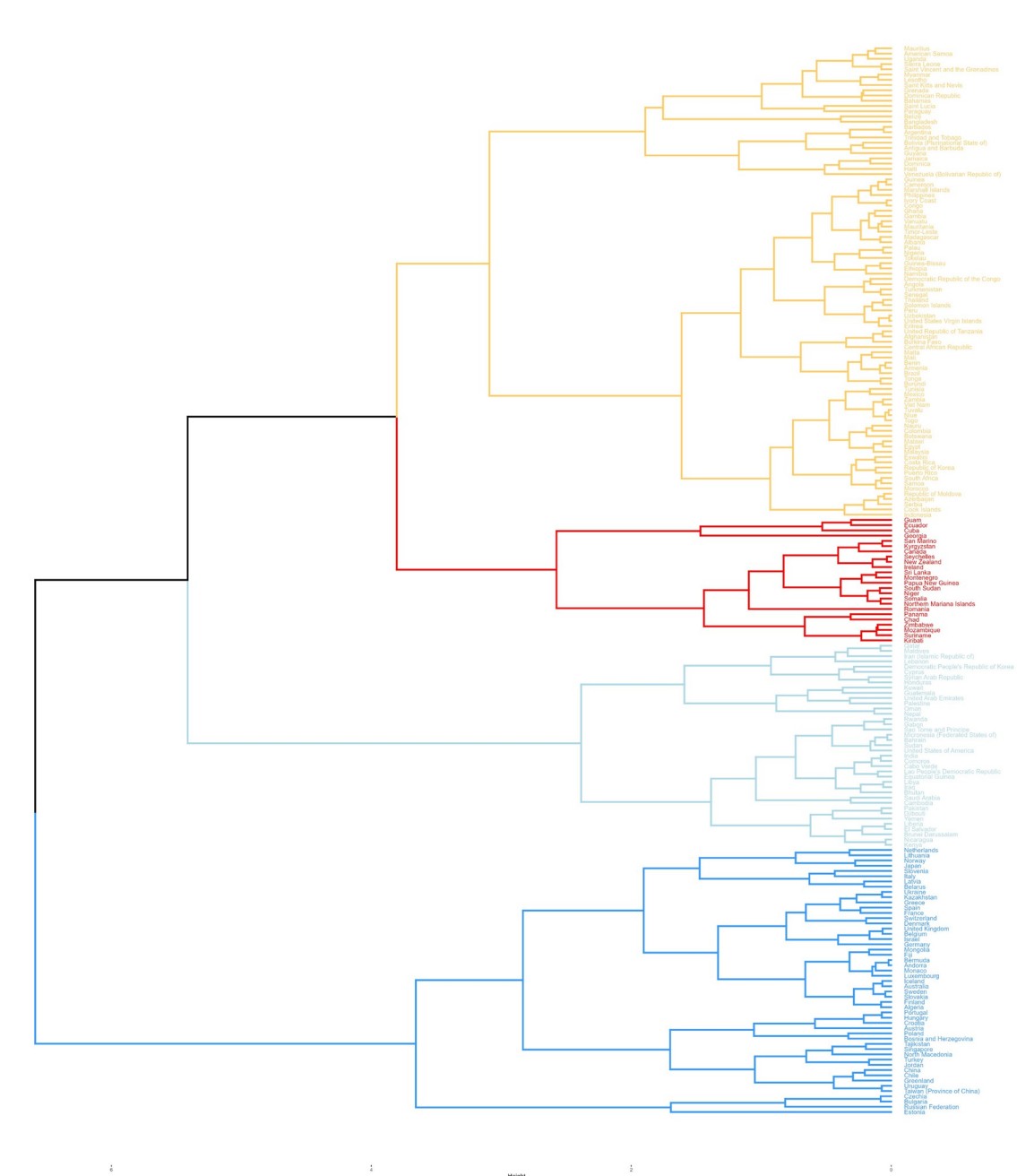

**Fig 3. Cluster analysis of incidence rate and death rate in different countries.**

and countries. High-income North America, High-income Asia Pacific, and Australasia closely followed expected trends over the study period. However, in many other regions, the actual situation is far from the forecast curve, for example, the level in regions such as Oceania and Southern Latin America is much higher than the expected level, however, the level in regions such as Tropical Latin America and Southern Sub-Saharan Africa is much lower than the expected level. The ASIR in most regions gradually declines smoothly along with the rise

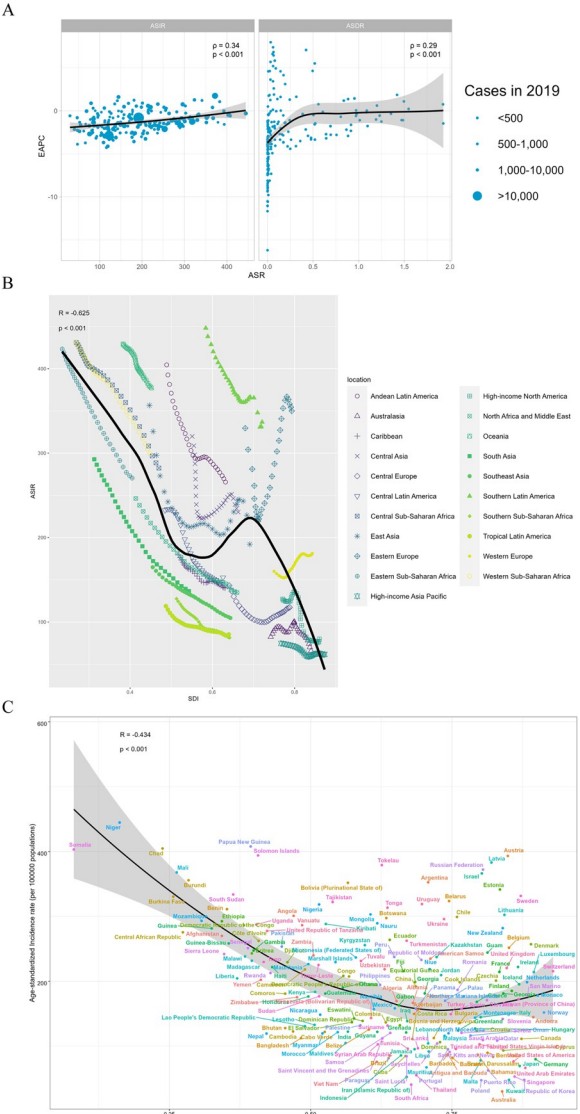

**Fig 4. Correlation analysis.** (A) The correlation between EAPC and ASIR/ASDR in 2019. (B) ASIR for EP for different regions and (C) countries and territories by SDI, 1990–2019.

in SDI, but there are some regions where the ASIR fluctuates sharply, such as Central Asia and Eastern Europe (Fig 4B).

At the overall level, ASDR is significantly negatively correlated with SDI values (R = -0.699, p < 0.001). Most of the high SDI regions have limited their ASDR to a very low level, close to zero, but almost all sub-Saharan Africa regions have higher than expected ASDR values. The case of AS-DALYs is basically a replica of the above (S4A and S4C Fig).

In 2019, there was an inverse relationship between the ASIR of EP and SDI at the national level, with some exceptions (Fig 4C). A similar pattern was observed in the relationship between ASDR and AS-DALYs in relationship to SDI. We can find that some countries are much higher than expected levels, such as Russian Federation, Austria, and Papua New Guinea, while countries at lower SDI, such as Somalia, Chad, and Niger, generally have higher

incidence rate, death rate and AS-DALYs, which is consistent with our findings mentioned above (S4B and S4D Fig).

### 3.5 EP-related risk factors

According to the GBD 2019, we only found one risk factor associated with EP, iron deficiency. Iron deficiency is classified as " Child and maternal malnutrition " and " Behavioral risks ". Globally, 21.9% of deaths and DALYs in patients with EP can be attributed to iron deficiency. Among the five SDI tiers, only the low SDI regions exceeded the global average for the proportion attributed to iron deficiency, reaching 22.8% and 22.9% for the proportion attributed to deaths and DALYs. We can also observe a gradual decrease in the contribution of iron deficiency to deaths and DALYs from low SDI regions to high SDI regions until the proportions reach a minimum of 11.1% and 10.8%, respectively. Regionally, the highest proportion of deaths and DALYs due to iron deficiency was found in sub-Saharan Africa and the lowest in Western Europe, which is consistent with our results above. It is worth noting that all Sub-Saharan Africa regions had more than 20% of EP deaths and DALYs attributable to iron deficiency. Only South Asia and Western Saharan Africa exceeded the global average for both indicators (Fig 5).

## 4. Discussion

EP is a leading cause of maternal morbidity and unexpected death worldwide. The incidence of EP is increasing in developed countries like the United States, while the mortality rate of EP in developing countries continues to rise, leading to a significant disease burden. Previous epidemiological studies have primarily focused on specific countries or regions, lacking a comprehensive global analysis to guide the prevention and treatment of EP [19]. Therefore, this study utilizes GBD 2019 data to provide a comprehensive overview of global epidemiological trends and patterns of EP over the past three decades. This analysis can contribute to the formulation of health policies and the effective allocation of healthcare resources.

EP is highly preventable and treatable, and with early detection, the chances of successful treatment are high, leading to a low risk of mortality. From a global perspective, there has been a decrease in the ASIR, ASDR, and AS-DALYs of EP in 2019 compared to 1990. This decreasing trend indicates improvements in the prevention and treatment of EP, which can be attributed to advancements in various aspects, including social living conditions. The age structure of EP incidence remains stable, with the highest incidence observed in the 25–29 years age group. Furthermore, the trends of mortality and DALYs rates with age are remarkably similar. This similarity may be due to the fact that DALYs in EP are primarily associated with fatal cases. However, it is surprising to note that both mortality and DALYs peak at an earlier age compared to 1990, suggesting a younger age profile of EP deaths. A survey conducted in Washington State, USA, between 1987 and 2014 revealed a decrease in EP hospitalization rates. However, there was an increase in EP-related mortality/severe morbidity among females aged 25–34 years, which aligns with the findings of our study. This outcome may be influenced by increasing low-risk EP patients receiving outpatient methotrexate therapy [20].

SDI represents a country or region's combined educational, economic, and medical levels. In our analysis of EP incidence rates, we observed a significant decline in EP incidence rates in regions with low SDI, low-middle SDI, and middle SDI. The rate of decline in these regions was higher than the global average. However, in regions with higher SDI, the rate of decline was relatively low. We hypothesize that this discrepancy may be because the ASIR in high SDI regions was already lower than the ASIR in low SDI regions in 2019, as early as 1990. Consequently, the potential for further decline was originally smaller, resulting in a less pronounced

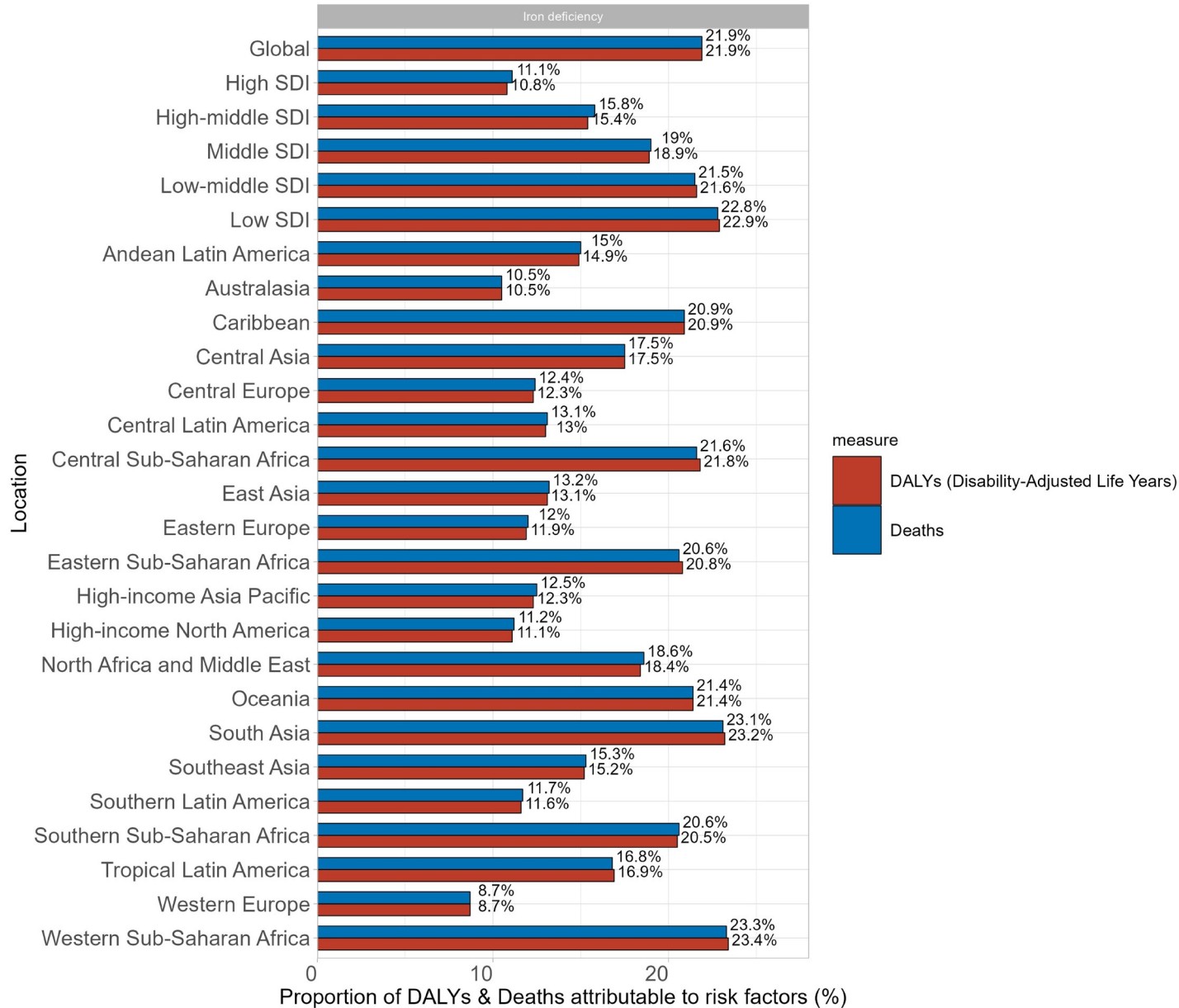

**Fig 5. Proportion of EP deaths and DALYs attributable to iron deficiency for different SDI regions, 2019.**

decreasing trend. Further research and investigation are necessary to better understand the underlying factors contributing to these trends.

Not only did the ASDR and AS-DALYs show varying degrees of decline in different SDI regions, but the correlation analysis also revealed a negative correlation between all three ASR indicators and SDI. Notably, Sub-Saharan African regions had higher than expected values of ASDR and AS-DALYs, indicating a concerning health situation for women in these low SDI regions. Iceland is a highly developed country, belonging to the High SDI countries, and a study showed that there were no maternal deaths from EP in Iceland from 1985–2009, which is inseparable from good living conditions, universal education levels, and a comprehensive national health care system. These factors play crucial roles in preventing and managing EP

incidences [21]. In contrast, low-income groups may face challenges such as lower levels of medical literacy and poorer living and medical conditions. Consequently, these factors could contribute to higher morbidity and mortality rates from EP in low SDI areas compared to high SDI areas [22].

Interestingly, when we analyze the data by age groups, we observe a delayed peak age of incidence in both High SDI areas and High-Middle SDI areas in 2019. In fact, in some age groups, the incidence rates have even exceeded the rates observed thirty years earlier. This delayed peak age of incidence and the increased incidence may be due to a variety of reasons. Firstly, women in higher SDI areas may have higher levels of education and contraceptive use. This may result in fewer early pregnancies and cases of younger EP, thus delaying the peak incidence of EP [23, 24]. Furthermore, it has been suggested that there is a negative correlation between the human development index and fertility rates. Therefore, we can hypothesize that the increased years of education in women may have led to a delay in the age of childbearing, consequently causing a delay in the onset of EP and an increased incidence at certain ages [25].

The EP situation in underdeveloped countries is concerning. The incidence of EP in underdeveloped countries such as Niger and Chad is still among the highest in the world due to a series of implication effects caused by economic backwardness. In addition, the low level of treatment leads to a high mortality rate of EP, whose DALYs can be hundreds or even thousands of times higher than those in developed countries. Previous research has highlighted significant disparities in the management of EP based on factors such as race, region, and access to reliable health insurance. For instance, some women may delay seeking medical care due to a lack of dependable health insurance, consequently increasing their risk of complications and death related to EP [26]. Furthermore, it has been observed that minority populations exhibit higher rates of reproductive tract infections, which can contribute to more severe cases of EP [22]. Addressing these disparities and closing the gap in EP management is challenging. It is influenced by long-standing economic and medical inequalities and limitations associated with factors like educational access and religious beliefs.

Previous studies have identified several common risk factors associated with EP, including reproductive mycoplasma infections, Chlamydia trachomatis infections, pelvic inflammatory disease, and the use of assisted reproductive technology and IUDs [27–29]. However, according to the GBD 2019, iron deficiency has emerged as a significant risk factor for EP. The study revealed that more than 20% of EP deaths and DALYs in sub-Saharan Africa were attributed to iron deficiency. Even in regions classified as having a High SDI, the proportion of EP-related deaths and DALYs associated with iron deficiency exceeded 10%. This finding offers a new direction for EP-related research. Furthermore, research has indicated that serum zinc levels are significantly higher in EP patients, while serum copper levels are lower. The copper/zinc ratio has shown potential as a novel diagnostic tool for EP. However, further basic research is needed to verify whether iron deficiency or other micronutrient deficiencies contribute to the development of EP.

The present study has several notable strengths. Firstly, it provides a comprehensive analysis of EP morbidity, mortality, and DALYs over a period of 30 years (1990–2019), offering a valuable overview of the global burden of EP. Additionally, the study examines the influencing factors associated with EP by comparing its conditions in different SDI and age groups, thereby providing new insights and directions for further research on EP. This study is the first-ever global epidemiological study on EP, making it an important reference for EP management. However, despite these strengths, there are some limitations that should be acknowledged. Firstly, the study relies on data from the GBD estimation, which may introduce uncertainty due to variations in data availability across countries and regions. Moreover, the lack of a comprehensive EP surveillance system in certain low and Middle SDI countries/

regions further contributes to potential data limitations. Secondly, the study does not differentiate between subtypes or pathological types of EP in the GBD analysis, hindering a more detailed disease analysis. Furthermore, it is worth noting that only one EP-related risk factor, iron deficiency, was included in the study, while common factors such as Chlamydia trachomatis infections and pelvic inflammatory disease were not identified as significant risk factors. This could be considered an incomplete and potentially biased representation of EP risk factors. However, modifying the availability of such data can be challenging.

## 5. Conclusion

Over the past 30 years, there has been a gradual decrease in the global morbidity, mortality, and DALYs associated with EP. However, it is crucial to note that these indicators are still increasing in economically disadvantaged regions. This trend may be attributed to various factors such as ethnicity, economic status, and educational levels. It is therefore essential to develop effective public health policies that address these disparities and provide enhanced protection for women belonging to ethnic minorities and low-income groups. Additionally, promoting early diagnosis and treatment of EP should be prioritized to mitigate its impact on these vulnerable populations.

## Supporting information

**S1 Table. Incidence of ectopic pregnancy in 1990 and 2019 for all locations, with EAPC from 1990 and 2019.**
(DOCX)

**S2 Table. DALYs of ectopic pregnancy in 1990 and 2019 for all locations, with EAPC from 1990 and 2019.**
(DOCX)

**S3 Table. Deaths of ectopic pregnancy in 1990 and 2019 for all locations, with EAPC from 1990 and 2019.**
(DOCX)

**S1 Fig. Distribution of case changes in various countries and regions in 2019.**
(TIF)

**S2 Fig. Distribution of DALYs rate in different SDI regions, age distribution of incidence rate in different SDI regions from 1990–2019.**
(TIF)

**S3 Fig. Distribution of death rate in different SDI regions, age distribution of incidence rate in different SDI regions from 1990–2019.**
(TIF)

**S4 Fig. Correlation analysis.** (A) ASDR for EP for different regions and (B) countries and territories by SDI, 1990–2019; (C)AS-DALYs for EP for different regions and (D) countries and territories by SDI, 1990–2019.
(TIF)

## Acknowledgments

The authors thank all staff for their contributions to the GBD database.

## Author Contributions

**Conceptualization:** Shufei Zhang.

**Funding acquisition:** Li Hong.

**Investigation:** Lian Yang, Hanyue Li, Jianming Tang.

**Visualization:** Shufei Zhang, Jianfeng Liu.

**Writing – original draft:** Shufei Zhang.

**Writing – review & editing:** Jianfeng Liu, Li Hong.

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
