## [Decision Letter · Decision Letter 0]

20 Apr 2023

PONE-D-22-31092Global burden and trends of ectopic pregnancy: An observational trend study from 1990 to 2019PLOS ONE

Dear Dr. Hong,

Thank you for submitting your manuscript to PLOS ONE. After careful consideration, we feel that it has merit but does not fully meet PLOS ONE’s publication criteria as it currently stands. Therefore, we invite you to submit a revised version of the manuscript that addresses the points raised during the review process.

ACADEMIC EDITOR: Please insert comments here and delete this placeholder text when finished. Be sure to:

I agree with the reviewer's comment.

We look forward to receiving your revised manuscript.

Kind regards,

Gang Qin, PhD, MD

Academic Editor

PLOS ONE

Journal Requirements:

"The National Key Research and Development Program of China (2018YFC2002204), Hubei Key Research and Development Program (2022BCA045) and The National Natural Science Foundation of China (81971364 & 82001527)."   

Additional Editor Comments:

I agree with the reviewer's comment.

Reviewers' comments:

Reviewer's Responses to Questions

**Comments to the Author**

1. Is the manuscript technically sound, and do the data support the conclusions?

Reviewer #1: Yes

2. Has the statistical analysis been performed appropriately and rigorously? 

Reviewer #1: Yes

3. Have the authors made all data underlying the findings in their manuscript fully available?

Reviewer #1: Yes

4. Is the manuscript presented in an intelligible fashion and written in standard English?

Reviewer #1: Yes

5. Review Comments to the Author

Reviewer #1: I appreciate the authors for this intensive research work on the topic “Global burden and trends of ectopic pregnancy: An observational trend study from 1990 to 2019”. The presentation of the data seems strong, and the paper is generally well-written, with an appropriate focus. However, this study may be accepted for publication with the following suggestions and comments for further improvements.

Abstract

Comment 1: The abstract is written with an appropriate focus.

Introduction

Comment 2: The introduction section is poorly written.

Comment 3: In addition to the current contents of the introduction section, add at least two paragraphs in the social context like.

1. What is the situation of EP in developed, developing, and underdeveloped countries? elaborate.

2. How is it changing over time?

3. Where it is still high from the previous literature.

4. Elaborate if it is declining, then why is your study important?

Material and Methods

Comment 4: Written Properly.

Results and Discussion

Comment 5: Figure resolution is very poor. Please provide high-resolution figures.

Comment 6: Add some discussion for underdeveloped and developing countries’ situations.

Comment 7: The conclusion is written very well.

6. PLOS authors have the option to publish the peer review history of their article (what does this mean?). If published, this will include your full peer review and any attached files.

Reviewer #1: **Yes: **Mayank Singh

---

## [Author Response · Author response to Decision Letter 0]

22 Apr 2023

Dear Editor and reviewer, 

Thank you for your letter and for the reviewers' comments concerning our manuscript entitled " Global burden and trends of ectopic pregnancy: An observational trend study from 1990 to 2019" (PONE-D-22-31092). Those comments are all valuable and very helpful for revising and improving our paper, as well as the important guiding significance to our researches. Images cannot be uploaded in this section of the submission system, so we have also uploaded a word version including the illustrations, which you can see at the end of the PDF.

First, here is our response to the Journal Requirements.

Response:

First of all, we would like to thank the journal again for their guidance, and we have reworked the manuscript style as requested by the journal.

2. Thank you for stating the following financial disclosure: "The National Key Research and Development Program of China (2018YFC2002204), Hubei Key Research and Development Program (2022BCA045) and The National Natural Science Foundation of China (81971364 & 82001527)." 

Response:

The above funds are our funders, but "The funders had no role in study design, data collection and analysis, decision to publish, or preparation of the manuscript. " And we will explain this again in the cover letter.

Response:

We double-checked our account and found that the corresponding author's ORCID iD exists in the system, as shown below. 

Response:

We have added the title of the supporting information at the end of the manuscript and updated the references in the manuscript accordingly.

Response:

Thank you for your reminder that we take copyright issues very seriously, so we immediately went to investigate the source of the map. Since our map was generated by R, we investigated the source of the map in the R package (as shown below) and found that it came from Natural Earth (http://www.naturalearthdata.com/), which is in the public domain and therefore does not require special attribution, but we will make the source of the Figure clear in the manuscript.

Response:

We have checked the reference list, and we can now be sure that it is correct.

Here is a point-by-point response to the comments and concerns of the reviewers.

Comment 1: The abstract is written with an appropriate focus.

Comment 4: Written Properly.

Comment 7: The conclusion is written very well. 

Response:

Thank you very much for your recognition of our work, and I believe that with your suggestions, our article will be more rigorous and more valuable.

Comment 2: The introduction section is poorly written. 

Response:

We are very sorry for the poorly written introduction, we have revised the manuscript according to your comments and added the key missing parts you suggested, thank you very much for your pointers.

Comment 3: In addition to the current contents of the introduction section, add at least two paragraphs in the social context like. 

1. What is the situation of EP in developed, developing, and underdeveloped countries? elaborate.

2. How is it changing over time?

3. Where it is still high from the previous literature.

4. Elaborate if it is declining, then why is your study important?

Response:

Your suggestions were meticulous and thoughtful, and we have added to the content of the manuscript.

Comment 5: Figure resolution is very poor. Please provide high-resolution figures.

Response:

We are very sorry that it affects the readability of the article. The images we uploaded in the system are very clear, but the image quality is compressed when the system automatically generates PDF files, so you can download the original images we uploaded by clicking the download button on the top right corner of the image (as shown in the picture), thank you very much for your patience. 

Comment 6: Add some discussion for underdeveloped and developing countries’ situations.

Response:

As you suggested we have added a discussion of underdeveloped and developing countries to the manuscript.

We are very grateful to the editor and reviewers for giving us this opportunity to revise the manuscript. We have tried our best to improve the manuscript, made some changes to the manuscript, and responded to the reviewers' questions one by one. We have uploaded the Manuscript and Revised Manuscript with Track Changes. We appreciate for editor and reviewers' warm work earnestly, and hope that the correction will meet with approval. Once again, thank you very much for your comments and suggestions. 

Thank you and best regards.

Yours sincerely,

Shufei Zhang 

Corresponding author：

Li Hong, Ph.D.

E-mail: dr_hongli@whu.edu.cn

---

## [Decision Letter · Decision Letter 1]

3 Jul 2023

PONE-D-22-31092R1Global burden and trends of ectopic pregnancy: An observational trend study from 1990 to 2019PLOS ONE

Dear Dr. Hong,

Thank you for submitting your manuscript to PLOS ONE. After careful consideration, we feel that it has merit but does not fully meet PLOS ONE’s publication criteria as it currently stands. Therefore, we invite you to submit a revised version of the manuscript that addresses the points raised during the review process.

ACADEMIC EDITOR: Pls revise according to the reviewer's suggestions.

We look forward to receiving your revised manuscript.

Kind regards,

Gang Qin, PhD, MD

Academic Editor

PLOS ONE

Journal Requirements:

Reviewers' comments:

Reviewer's Responses to Questions

**Comments to the Author**

1. If the authors have adequately addressed your comments raised in a previous round of review and you feel that this manuscript is now acceptable for publication, you may indicate that here to bypass the “Comments to the Author” section, enter your conflict of interest statement in the “Confidential to Editor” section, and submit your "Accept" recommendation.

Reviewer #2: (No Response)

2. Is the manuscript technically sound, and do the data support the conclusions?

Reviewer #2: Partly

3. Has the statistical analysis been performed appropriately and rigorously? 

Reviewer #2: No

4. Have the authors made all data underlying the findings in their manuscript fully available?

Reviewer #2: Yes

5. Is the manuscript presented in an intelligible fashion and written in standard English?

Reviewer #2: No

6. Review Comments to the Author

Reviewer #2: Introduction:

1. The introduction section is poorly written

2. There is no need to discuss diagnosis and treatment in the introduction

Methods

1. Study data was obtained from The Global Burden of Disease Study 2019 was modeled by by the Institute for Health Metrics and Evaluation (IHME). SDI data obtained from Global Health Data Exchange (GHDx)

2. Global Burden of Disease Study 2019 should be introduced in details

3. SDI classes should be explained

4. How is EP defined in GED protocols?

5. Age-standardized incidence rate, mortality rate and DALY rate were defined in GBD, why haven't researchers used them and recalculated?

6. “EAPCs were calculated using a linear regression model as follows: ln (ASR) = α + β x + ε,” is incorrect

7. R language version 4.2.1 is incorrect, R software is correct

Results

1. 3.1 Distribution and trends in the incidence rate of EP by age or year???

2. Figures resolution is very poor. Please provide high-resolution figures

3. Figure 2 is unclear

4. the 0-14 and e 55+ years age groups should be excluded

5. the results of EP-related risk factors not well presented

discussion

1. discussion section is weak

2. advantage and limitation were missed

7. PLOS authors have the option to publish the peer review history of their article (what does this mean?). If published, this will include your full peer review and any attached files.

Reviewer #2: **Yes: **Fatemeh khosravi Shadmani

---

## [Author Response · Author response to Decision Letter 1]

17 Jul 2023

Dear Editors and Reviewers,

Thank you for your letter and for the reviewers' comments concerning our manuscript entitled " Global burden and trends of ectopic pregnancy: An observational trend study from 1990 to 2019" (PONE-D-22-31092R1). Those comments are all valuable and very helpful for revising and improving our paper, as well as the important guiding significance to our research. 

First, here is our response to the Journal Requirements.

Response:

Thank you for your patience in reviewing the manuscript, we have checked all the references to ensure that none of them have been withdrawn, and although we have changed some references, we can be sure that the reference list is correct.

Here are our responses to the reviewers' general comments

We truly appreciate your suggestions on our manuscript and are very sorry that the article's readability has been reduced due to our writing, so we have tried our best to polish the language in the revised manuscript. As for the statistical issues, we have also made changes and deletions based on your suggestions. We hope the revised manuscript could be acceptable to you.

Here is a point-by-point response to the comments and concerns of the reviewers.

Comments on the introduction:

1. The introduction section is poorly written

Response:

We sincerely apologize for our poor writing, and we have carefully and completely rewritten the Introduction section of the manuscript.

2. There is no need to discuss diagnosis and treatment in the introduction

Response:

Thank you very much for your suggestions, we couldn't agree with you more, so we have removed the relevant content and have carefully rewritten this section, and we hope that these changes will fulfill your requirements for the manuscript.

Comments on the Methods:

1. Study data was obtained from The Global Burden of Disease Study 2019 was modeled by the Institute for Health Metrics and Evaluation (IHME). SDI data obtained from Global Health Data Exchange (GHDx)

Response:

First of all, thank you very much for your suggestions, we have added this important information to the revised manuscript.

Line 62-64, “Study data was obtained from GBD 2019 was modeled by the Institute for Health Metrics and Evaluation (IHME)” was added.

Line 70-71, “SDI data was obtained from Global Health Data Exchange (GHDx)” was added.

2. Global Burden of Disease Study 2019 should be introduced in details

Response:

We are very sorry for our omission of this important section, so we have added an introduction to the Global Burden of Disease Study 2019 in the revised manuscript.

Line 53-55, “Explore results from the 2019 Global Burden of Disease study (GBD 2019), which published epidemiological data related to 369 diseases/injuries and 286 causes of death, covered EP in 204 countries and territories from 1990 to 2019” was added.

References:

http://doi.org/10.1016/j.ajp.2023.103677

3. SDI classes should be explained

Response:

We are very sorry that our description of SDI was not clear, so we have made it explicit in the revised manuscript.

Line 65-69, “The Socio-demographic Index (SDI) is a composite indicator of development status strongly correlated with health outcomes. It is the geometric mean of 0 to 1 indices of total fertility rate under the age of 25, mean education for those ages 15 and older and lag distributed income per capita. As a composite, a location with an SDI of 0 would have a theoretical minimum level of development relevant to health, while a location with an SDI of 1 would have a theoretical maximum level” was added.

References:

https://ghdx.healthdata.org/sites/default/files/record-attached-files/IHME_GBD_2019_SDI_1950_2019_INFO_SHEET_Y2021M08D16.PDF

4. How is EP defined in GBD protocols?

Response:

Thank you very much for your suggestion regarding the definition of ectopic pregnancy, which is a very important issue. We have managed to find the GBD definition of ectopic pregnancy after a careful search of the sources and have added it to the revised manuscript.

Line 61, “In GBD, EP is defined as pregnancy occurring outside of the uterus.” was added.

References:

https://www.healthdata.org/results/gbd_summaries/2019/ectopic-pregnancy-level-4-cause

5. Age-standardized incidence rate, mortality rate and DALY rate were defined in GBD, why haven't researchers used them and recalculated?

Response:

Thank you very much for your suggestion, we have calculated the Age-standardized incidence rate, mortality rate, and DALY rate in the table and abbreviated them as ASIR, ASDR, and AS-DALYs in the Method section and have used them for calculation and visualization in the results. 

6. “EAPCs were calculated using a linear regression model as follows: ln (ASR) = α + β x + ε,” is incorrect

Response:

Thank you for your correction, we have corrected it in the revised manuscript.

Line 77-78, “EAPCs were calculated using a linear regression model as follows: ln (ASR) = α + β x + ε” was corrected as “The EAPC is calculated by fitting the linear regression line: Y = α + βx + ε, where y represents ln(ASR) and x refers to the calendar year.”.

7. R language version 4.2.1 is incorrect, R software is correct

Response:

Thanks again for the correction; we have also corrected it in the revised manuscript.

Line 87, “R language” was corrected as “R software”.

Comments on the Results:

1. 3.1 Distribution and trends in the incidence rate of EP by age or year???

Response:

Thank you very much for your patience, we have corrected several titles that did not make sense in the revised manuscript.

Line 91, “3.1 Distribution and trends in the incidence rate of EP by age or year” was corrected as “3.1 Distribution and trends in the incidence rate of EP”.

Line 115, “3.2 Distribution and trends in the DALYs rate of EP by age or year” was corrected as “3.2 Distribution and trends in the DALYs rate of EP”.

Line 133, “3.3 Distribution and trends in the Death rate of EP by age or year” was corrected as “3.3 Distribution and trends in the Death rate of EP”.

2. Figures resolution is very poor. Please provide high-resolution figures

Response:

We are very sorry that the unclear figures have caused problems for your review, we have remade and uploaded all the figures and ensured their clarity. However, the journal requires that figures should not exceed 10M in size, so there may be some figures with crowded and unclear details.

3. Figure 2 is unclear

Response:

Thanks for your suggestion, we have recreated and uploaded a clear figure 2.

4. the 0-14 and e 55+ years age groups should be excluded

Response:

We acknowledge that it may be unreasonable to discuss ectopic pregnancies in women under 14 years old or over 55 years old, therefore, we excluded cases in these two age groups and re-visualized them, as shown in Figure 2, S4 Figure, and S5 Figure. 

5. the results of EP-related risk factors not well presented

Response:

Thank you for your suggestion and we recognize the limitations of this study in terms of the risk factors associated with ectopic pregnancy. This is because GBD 2019 contains only one risk factor for ectopic pregnancy, iron deficiency, and therefore we have only explored it in the manuscript. We have also rewritten this section in the revised manuscript, and we hope that these changes will help the reader better understand the contents of the manuscript.

Comments on the discussion:

1. discussion section is weak

Response:

Thank you very much for your patient suggestions, we have carefully rewritten the discussion section of the manuscript. The revised manuscript provides a comprehensive analysis of the condition of ectopic pregnancy on several levels.

2. advantage and limitation were missed

Response:

We recognize that an exploration of the advantages and limitations of this study was missing from our manuscript, so in the revised manuscript, we have added this section. Thanks to your suggestions, this makes our manuscript better.

Line 271-286, “The present study has several notable strengths. Firstly, it provides a comprehensive analysis of EP morbidity, mortality, and DALYs over a period of 30 years (1990-2019), offering a valuable overview of the global burden of EP. Additionally, the study examines the influencing factors associated with EP by comparing its conditions in different SDI and age groups, thereby providing new insights and directions for further research on EP. This study is the first-ever global epidemiological study on EP, making it an important reference for EP management. However, despite these strengths, there are some limitations that should be acknowledged. Firstly, the study relies on data from the GBD estimation, which may introduce uncertainty due to variations in data availability across countries and regions. Moreover, the lack of a comprehensive EP surveillance system in certain low and Middle SDI countries/regions further contributes to potential data limitations. Secondly, the study does not differentiate between subtypes or pathological types of EP in the GBD analysis, hindering a more detailed disease analysis. Furthermore, it is worth noting that only one EP-related risk factor, iron deficiency, was included in the study, while common factors such as Chlamydia trachomatis infections and pelvic inflammatory disease were not identified as significant risk factors. This could be considered an incomplete and potentially biased representation of EP risk factors. However, modifying the availability of such data can be challenging.” was added.

We are very grateful to the editor and reviewers for giving us this opportunity to revise the manuscript. We have tried our best to improve the manuscript, made some changes to the manuscript, and responded to the reviewers' questions one by one. We have uploaded the Manuscript and Revised Manuscript with Track Changes. We appreciate for editor and reviewers' warm work earnestly and hope that the correction will meet with approval. Once again, thank you very much for your comments and suggestions. 

Thank you and best regards.

Yours sincerely,

Shufei Zhang 

Corresponding author：

Li Hong, Ph.D.

E-mail: dr_hongli@whu.edu.cn

---

## [Editor Report · Decision Letter 2]

29 Aug 2023

Global burden and trends of ectopic pregnancy: An observational trend study from 1990 to 2019

PONE-D-22-31092R2

Dear Dr. Hong,

We’re pleased to inform you that your manuscript has been judged scientifically suitable for publication and will be formally accepted for publication once it meets all outstanding technical requirements.

Kind regards,

Gang Qin, PhD, MD

Academic Editor

PLOS ONE
---

## [Editor Report · Acceptance letter]

31 Aug 2023

PONE-D-22-31092R2 

Global burden and trends of ectopic pregnancy: An observational trend study from 1990 to 2019 

Dear Dr. Hong:

I'm pleased to inform you that your manuscript has been deemed suitable for publication in PLOS ONE. Congratulations! Your manuscript is now with our production department. 

Kind regards, 

on behalf of

Dr. Gang Qin 

Academic Editor

PLOS ONE